# P2P: Transforming from Point Supervision to Explicit Visual Prompt for Object Detection and Segmentation

## Abstract

Point-supervised vision tasks, including detection and segmentation, aiming to learn a network that transforms from points to pseudo labels, have attracted much attention in recent years. However, the lack of precise object size and boundary annotations in the point-supervised condition results in a large performance gap between point- and fully-supervised methods. In this paper, we propose a novel iterative learning framework, Point to Prompt (P2P), for point-supervised object detection and segmentation, with the key insight of transforming from point supervision to explicit visual prompt of the foundation model. The P2P is formulated as an iterative refinement process of two stages: Semantic Explicit Prompt Generation (SEPG) and Prompt Guided Spatial Refinement (PGSR). Specifically, SEPG serves as a prompt generator for generating semantic-explicit prompts from point input. In PGSR stage, prompts guide the visual foundation model to further refine the object regions, by leveraging the outstanding generalization ability of the foundation model. The two stages are iterated multiple times to improve the quality of predictions progressively. Experimental results on multiple datasets demonstrate that P2P achieves SOTA performance in both detection and segmentation tasks, further narrowing the performance gap with fully-supervised methods.

## 1 Introduction

In recent years, significant progress has been made in the field of computer vision, thanks to the efficient backbone network (Simonyan & Zisserman, 2014; He et al., 2016; Dosovitskiy et al., 2020; Liu et al., 2021) and the support of large-scale manually annotated datasets (Lin et al., 2014; Everingham et al., 2015; Russakovsky et al., 2015). However, the process of manually annotating a large-scale dataset with detailed annotations (*e.g.*, precise bounding boxes or pixel-level masks) is expensive and requires a substantial amount of human effort.

To solve this problem, weakly supervised visual understanding tasks, such as weakly supervised object detection and segmentation have gained widespread attention. Typically, the weak supervision includes image-level (Bilen & Vedaldi, 2016; Wan et al., 2019; Xu et al., 2022; Zhou et al., 2018), point-level (Chen et al., 2022; Liao et al., 2023; Gao et al., 2022) and scribble-level (Zhang et al., 2020; He et al., 2023a), *etc.*. Among them, object detection and segmentation with point-level supervision (shortened to PSOD and PSOS, respectively) have attracted the growing attention in recent years thanks to the low annotation burden and distinctive location information of points.

However, the performance of existing point-supervised methods is still far from satisfactory, about 59% and 55% of that of fully supervised detection and segmentation baselines. This is contrary to the core of weakly supervised methods, *i.e.*, ***to release the annotation burden while still achieving decent performance.*** PSOD is used as an example to illustrate the limitations of point-supervised vision tasks. The current PSOD methods generally follow the paradigm of Multiple Instance Learning (MIL) or Cascaded MIL (C-MIL) fashion. They first use proposal generation methods (*e.g.*, Selective Search (Uijlings et al., 2013), MCG (Arbeláez et al., 2014) or neighbor proposal sampling (Chen et al., 2022)) to construct proposal bags. After that, top-$k$ proposals with high scores are selected from hundreds of independent proposals as the final result by MIL. Due to the lack of object

Figure 1: Training paradigms with two different PSOD frameworks: (a) Basic PSOD framework, generally using a cascaded MIL fashion. (b) Our P2P framework.

size and edge information, proposal generation is more random and low-quality, and the results are limited by inaccurate proposals. Additionally, multiple independent proposals bring high randomness in the selection process, and it is easy to converge to the sub-optimal solution and focus on the discriminative part rather than he entirety of the object.

Thanks to the substantial progress in the visual foundation models, such as Segment Anything Model (SAM) (Kirillov et al., 2023), many downstream tasks have witnessed significant breakthroughs (Wang et al., 2023b). *We hold the perspective that, rather than directly designing large foundation models, it is more meaningful to leverage them for specific tasks in resource-constrained situations.* Notably, some efforts have been made to adapt SAM for weakly supervised tasks, *e.g.*, weakly supervised semantic segmentation (Chen et al., 2023b; Sun et al., 2023). However, these studies have employed SAM as a supplementary tool in a simplistic way and have not attempted to explore how to better guide SAM by enhancing the semantic representation capability of prompts. Compared with bounding boxes or masks, point annotations inherently possess limited semantic representation. When points are directly used as prompts for SAM, only 40% of the masks cover more than 70% of foreground pixels, significantly lower than the results (about 80%) obtained when using boxes as prompts. This highlights the crucial importance of semantic-explicit prompts.

In this paper, we propose a novel framework, referred to as **P**oint-to-**P**rompt (P2P) for point-supervised detection and segmentation, by transforming the point supervision into visual prompt learning. To the best of our knowledge, this is the first work that attempts to switch point-supervised tasks into visual prompt learning. An overview of the contrast between the existing PSOD framework and our framework is presented in Fig. 1. P2P comprises two integral processes: the Semantic-Explicit Prompt Generation (SEPG) stage and the Prompt Guided Spatial Refinement (PGSR) stage. Specifically, the SEPG stage is designed to generate semantic explicit pseudo boxes as prompts under the guidance of semantic confidence. The PGSR stage further refines the target regions covered in the semantic-explicit prompts by leveraging the outstanding generalization ability of the foundation model SAM. It operates through an iterative process, involving multiple iterations between SEPG and PGSR, ultimately resulting in the generation of precise pseudo-labels. Utilizing SAM, our method can output precise pseudo-masks, so it can be applied as both a point-supervised detection and segmentation method that transforms points into accurate pseudo masks and boxes.

Experiments on the challenging MS COCO 2017 and PASCAL VOC 2007 datasets are conducted to validate both the detection and the segmentation performance. Given point supervision, P2P further closes the gap with the fully supervised model and achieves 84% and 75% of the performance of fully supervised on the COCO dataset, respectively. It outperforms the previous point-based detection and segmentation methods by a large margin. Our main contributions are as follows:

• We design a novel Point-to-Prompt (P2P) method for point-supervised object detection and segmentation, which transforms the point supervision into prompting to predict precise pseudo-labels.

• We propose an iterative learning framework using visual foundation model to achieve a semantic-explicit output, including a semantic-explicit prompt generation stage and a prompt guided spatial refinement stage.

• Our P2P method achieves state-of-the-art performance on detection and segmentation, significantly narrows the performance gap with fully supervised methods, and provides new insights for point supervision tasks.

## 2 RELATED WORKS

**Weakly Supervised Detection.** For image-supervised detection, the difficulty lies in how to mine the location of each instance with only semantic information. Existing methods (Bilen & Vedaldi, 2016; Tang et al., 2017; 2018; Wan et al., 2019; 2018; Gao et al., 2019; Chen et al., 2020) generally build image-level proposal bags containing hundreds of proposals, and then mine instance-level supervision through MIL, unsupervised clustering, and comparative learning. Due to the lack of location information, the performance is still poor for some complex datasets, such as COCO (Lin et al., 2014). Point-supervised methods benefit from additional point-level annotations, providing coarse location information. The difficulty of the point-level approach is how to search the size and boundary of the object so as to accurately estimate the bounding box. Ren et al. (2020b) designs a unified network compatible with various supervision forms. P2BNet (Chen et al., 2022) specifies that low-quality proposals limit the performance of point-supervised methods. It proposes to generate proposals through a neighbor sampling policy and designs a cascade MIL framework. However, the quality of proposals remains sub-optimal and a significant number of independent proposals introduce a degree of randomness into the learning process. Furthermore, existing methods are still constrained by the MIL paradigm.

**Weakly Supervised Segmentation.** Weakly supervised segmentation (primarily focused on instance segmentation) is mainly performed by estimating instance-level pseudo masks and refining the estimated masks by training a segmentation model. To obtain the pseudo mask for each instance with only point-level information, previous approaches have either used off-the-shelf proposal methods (Ahn et al., 2019) or generated instance-level localization maps (Kim et al., 2022; Liao et al., 2023) by refining the attention maps of CAM (Zhou et al., 2016) or ViT (Dosovitskiy et al., 2020). The performance of the current methods is significantly constrained by the quality of the attention map. Liao et al. (2023) analyzes that only about 30% of the ViT attention maps can cover more than 50% of the foreground objects, greatly limiting the performance.

**Prompting and Foundation Models.** Prompting refers to the process of designing prompts that enable the model to adapt and generalize to different tasks. By carefully designing task-relevant prompts, large foundation models can deliver superior performance on a variety of downstream tasks. Segment Anything Model (SAM) (Kirillov et al., 2023), as a representative prompt-based foundation model in CV, is designed for image segmentation and has brought a new trend in solving other downstream tasks. Considering the foundation model has been trained on large-scale data, prompting usually contributes to better model generalization on the downstream tasks, especially in the case of limited data or annotations. The research community has been actively engaged in exploring and pushing the capability boundaries of SAM and applying it to various tasks through effective prompts, *e.g.*, Remote Sensing (Chen et al., 2023a; Wang et al., 2023a), medical image analysis (He et al., 2023b; Ma & Wang, 2023), video object tracking (Yang et al., 2023), and weakly supervised semantic segmentation (Chen et al., 2023b; Sun et al., 2023; Jiang & Yang, 2023). Wang et al. (2023a) explores the suitability and effectiveness of bounding boxes in designing prompts for efficient annotation purposes. Inspired by these approaches, we apply prompting to point-supervised detection and segmentation tasks and propose converting points into prompts to facilitate the performance of point-supervised tasks.

## 3 METHODOLOGY

### 3.1 OVERVIEW

**Problem.** A point annotation $p$ can be represented as $p = (p_x, p_y, c)$, where $(p_x, p_y)$ and $c$ represent point location and object category, respectively. Point-supervised tasks aim to train a point-to-label regressor using point annotations to predict pseudo-labels. Subsequently, a task-related sub-network (*e.g.*, a detector) is optimized in a fully-supervised manner for inference. Thus, the core of this task lies in designing an accurate point-to-label regressor, denoted as $\Phi_{reg}(\cdot)$, which transforms the point annotation into precise pseudo annotations. To design a well-formed regressor, we introduce the P2P framework, proposing to first transform point supervision into explicit visual prompts and then obtain pseudo-labels guided by these prompts.

**Framework.** We structure the P2P as an iterative refinement process of the Semantic-Explicit Prompt Generation (SEPG) stage and the Prompt Guided Spatial Refinement (PGSR) stage. The

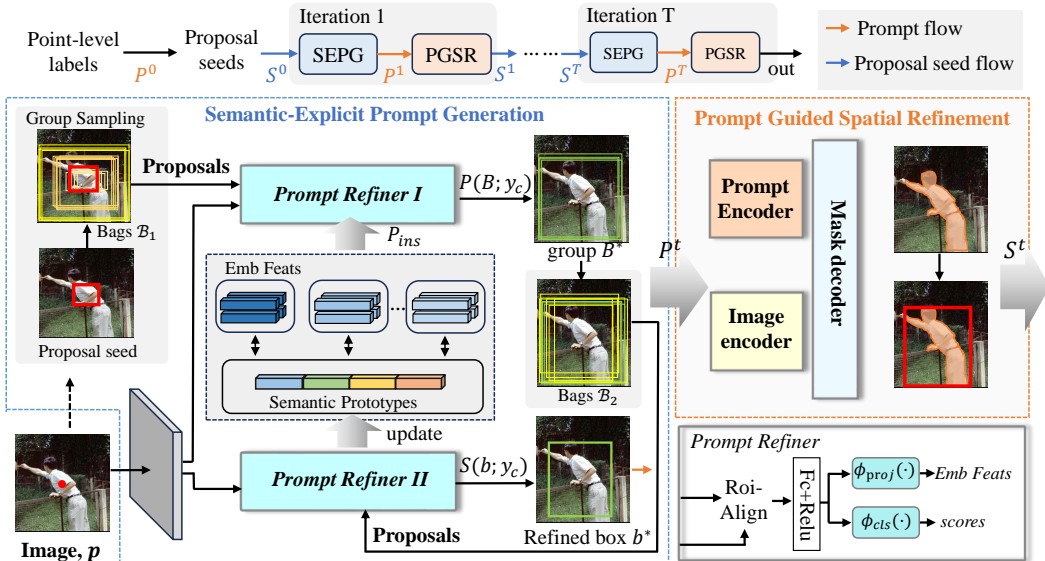

Figure 2: Framework of P2P, which performs SEPG and PGSR iteratively to generate more accurate pseudo labels. Specifically, given an image and point annotation, we first use SAM to generate proposal seeds as the initial input to SEPG. In SEPG, we use a group sampling strategy and two joint confidence-based refiners to get the refined box $b^*$. Guided by $b^*$, PGSR further spatially refines the object region to get more accurate masks and boxes under the given semantic and updates the proposal seeds.

two stages respectively serve the roles of "point to prompt" and "prompt to pseudo-mask". In P2P, the point annotation $p$ is first viewed as the prompt (denoted as $P^0$) of the foundation model to generate an initial mask for each object. The outer rectangle of the mask is used as the proposal seed $S^0$. Taking $S^0$ as input, P2P initiates the first round of iteration. While proposal seeds may not be entirely accurate, they can still provide valuable prior information on object size. Then, taking the initial proposal seeds $S^0$ as input, refined box is generated by two prompt refiners under semantic supervision. Compared to $S^0$, the refined box covers the main semantic part of the object and can be used as the subsequent prompt, refferd to as $P^1$. After, PGSR plays the role of spatial refinement guided by the semantic-explicit prompt $P^1$ and generates the next round of proposal seed $S^1$. Improved proposal seeds lead to better prompts, and better prompts, in turn, contribute to better proposal seeds. The two modules iterate $T$ times, ultimately yielding predicted pseudo-masks and pseudo-boxes through PGSR. The overall pipeline of P2P is depicted in Fig. 2.

## 3.2 SEMANTIC-EXPLICIT PROMPT GENERATION

We design a semantic prompt generator that takes semantic-agnostic seeds as input and produces semantic-explicit prompts for SAM. Our approach adopts a *group-then-individual* strategy on two refiners, as illustrated in Fig. 2. Initially, in *Prompt Refiner I*, we obtain a semantically accurate proposal group $B^*$, followed by *Prompt Refiner II*, we further refine the proposal group to obtain refined proposal $b^*$.

**Seeds-based Group Sampling.** Previous methods usually use neighbor sampling to build proposal bags that contain hundreds of individual proposals that usually suffer from low quality and lack of good priority. To mitigate that, we introduce a group sampling strategy based on the initial proposal seeds $S^0$. For the first phase, we create the proposal bag $\mathcal{B}_1$ by progressively sampling $n$ proposal groups $\{B_i\}_{i=1}^n$ for each instance based on the proposal seeds, *i.e.*, $\mathcal{B}_1 = \{B_i\}_{i=1}^n$. These distinct proposal groups are generated by scaling the proposal seeds at various scales. Each proposal group comprises $m$ proposals with strong spatial correlation, denoted as $B_i = \{b_{i,j}\}_{j=1}^m$, where $b_{i,j}$ denotes the $j$th proposal of the $i$th group and $m$ signifies the number of proposals in $B_i$. The concept behind designing proposal groups is to reduce the solution space by ***selecting proposal***

***group instead of individual proposals***. For the second phase, we construct the proposal bag $\mathcal{B}_2$ by augmenting the group of proposals produced in the first phase. We adopt a "proposal jittering" strategy to generate randomly jittered proposals in four orientations. Further details can be found in the Appendix.

**Proposal-to-Prompt Semantic Lifting.** Different from classical MIL frameworks, we adopt a *group-then-individual* strategy, *i.e.*, selecting a group of proposals with strong spatial correlation first and then further refining the proposals according to the group in the second phase. We employ a more stable feature prototype for computing group-based semantic distribution. The refiners in P2P comprise a classification head and an embedding head, which calculate classification scores and feature embeddings, respectively.

In *Prompt Refiner I*, the problem lies in identifying a semantic-accurate ***proposal group***, which determines the direction of model optimization. The basic MIL head or classification head is frequently employed as the refiner, but the inherent randomness and instability in its training process easily lead the model towards sub-optimal solutions. To remedy the bias of predicted probabilities, we use a prototype representation to obtain stable group-based semantic distributions of proposals.

A memory buffer is established to keep a set of prototypes $\mathcal{V} = \{\mathcal{V}_c\}_{c=1}^C$ for each category, which preserves the semantic-explicit features. These prototypes are updated using the selected high-quality embedding features from *Prompt Refiner II* in each iteration , via the Exponential Moving Average (EMA) algorithm. After that, the ***group-based instance-level probability*** distribution $\mathcal{P}_{ins}(B_i; y_c)$ of each proposal group $B_i$ can be measured by the similarity between the feature embeddings of the proposal groups and their corresponding semantic prototypes.

$$\mathcal{P}_{ins}(B_i; y_c) = \frac{\exp(sim(Z_i, \mathcal{V}_c))}{\sum_i \exp(sim(Z_i, \mathcal{V}_c))}, \tag{1}$$

where $Z_i$ indicates the feature embedding of the proposal group $B_i$. It is calculated by averaging the feature embeddings of all the proposals in the group, as $Z_i = \frac{1}{|B_i|} \sum_j z_{i,j}$, where $z_{i,j}$ indicates $j$th proposal in $B_i$, and $|B_i|$ is the number of proposals. $sim(\cdot, \cdot)$ denotes the cosine similarity metric, utilized to quantify the similarity between the embedding features and the semantic prototypes. Furthermore, we calculate the ***group-based semantic-level probability*** $\mathcal{P}_{sem}(B_i; y_c)$ for each proposal group $B_i$. It is computed by the score of the proposal group, as

$$\mathcal{P}_{sem}(B_i; y_c) = \frac{\exp\{\frac{1}{|B_i|} \sum_{b_{i,j} \in B_i} O(b_{i,j}; y)\}}{\sum_i \sum_y \exp\{\frac{1}{|B_i|} \sum_{b_{i,j} \in B_i} O(b_{i,j}; y_c)\}}, \tag{2}$$

where $O(b_{i,j}; y_c)$ denotes the score of $j$th proposal in $B_i$. Finally, we define a ***group-based joint probability*** distribution that combines the semantic level and the instance level, termed $\mathcal{P}(B_i; y_c) = \mathcal{P}_{ins}(B_i; y_c) * \mathcal{P}_{sem}(B_i; y_c)$, which indicates the semantic probability of proposal group $B_i$ for a given semantic label $y_c \in \{y_1, y_2, ..., y_C\}$. In the learning procedure, based on the above definition, $\mathcal{P}(B_i; y_c)$ is applied to the refinement process, the corresponding loss function of the first phase (termed $\mathcal{L}_1$) is defined as:

$$\mathcal{L}_1 = -\sum_{c=1}^C y_c \log \sum_i \mathcal{P}(B_i; y_c) + (1 - y_c)\log(1 - \sum_i \mathcal{P}_{sem}(B_i; y_c)). \tag{3}$$

The proposal group that contains multiple semantic-explicit proposals with the highest semantic confidence is selected, termed $B^*$.

*Prompt Refiner II* performs further ***proposal refinement*** as well as ***prototype update*** with a similar structure as the first phase, *i.e.*, comprising a classification head and an embedding head. For further refinement, based on the proposal group $B^*$, the proposal bag $\mathcal{B}_2$ is constructed as the input of this phase. The proposals in $\mathcal{B}_2$ maintain strong spatial correlation and have identified the main semantic regions. So in this phase, we only employ the general classification head and adopted the focal loss (termed $\mathcal{L}_2$) (Lin et al., 2017) for further refinement. The proposals with the top-$k$ highest scores are weighted to obtain the final refined box $b^*$. More details can be found in the Appendix.

For prototype update, we treat the proposal score of this phase as an indicator. For example, for the proposal $b \in \mathcal{B}_2$, the corresponding embedding feature and score are denoted as $z$ and $s$, respectively.

During each training iteration, the embedded features whose corresponding score $s$ exceeds a certain threshold $\tau$ are selected, as

$$v_c = \begin{cases} z_{i,j}, & s(b_{i,j}, y_c) \geq \tau, \\ 0, & otherwise, \end{cases} \tag{4}$$

where $v_c$ denotes the local prototype of the current iteration for category $c$. And then the global semantic prototypes are updated with local prototypes via the EMA algorithm, as $\mathcal{V}_c = \alpha * \mathcal{V}_c + (1 - \alpha) * v_c$, where $\mathcal{V}_c$ denotes the semantic prototype of category $c$, and $\alpha$ is momentum parameter and empirically set to be 0.99. Consequently, we obtain a set of prototypes $\mathcal{V} = \{\mathcal{V}_1, \mathcal{V}_2, ..., \mathcal{V}_C\}$ for all categories, which are continuously updated during the training process.

**Discussion.** The design of prototypes allows the two refiners to mutually reinforce each other. We use the scores from *Prompt Refiner II* as an indicator to update the semantic prototypes with high-quality embeddings. High-quality prototypes are then utilized in *Prompt Refiner I* to compute instance probabilities, yielding high-quality proposals. When these proposals are inputed into *Prompt Refiner II*, the quality of the semantic prototypes is further enhanced, leading to a mutually improving process.

### 3.3 PROMPT GUIDED SPATIAL REFINEMENT

In this stage, we mainly leverage the visual foundation model SAM to further refine the spatial regions of objects. Since SAM does not have the ability of semantic understanding, the ability of getting the desired output relies on the accuracy of the semantic prompts. The semantics of a point is indeterminate because an object contains multiple parts that represent different semantics (*e.g.* a *dress* and a *bag*), and there are semantic biases between parts and the global (*e.g.*, *clothes* and *person*). In contrast, refined boxes generated in the SEPG stage cover the main semantic regions of objects and serve as semantic explicit prompts to guide SAM in generating spatially refined regions.

SAM consists of three main components, an image encoder ($\Phi_{\text{img-e}}$), a prompt encoder ($\Phi_{\text{pmt-e}}$), and a mask decoder ($\Phi_{\text{mask-d}}$). In our approach, we input the boxes as prompts and the original point annotations as complementary prompts into SAM. The original image and the prompts are fed into the image encoder and the prompt encoder, respectively. After that, taking the intermediate features of image and prompts as input, the refined mask $\mathcal{M}$ is generated by $\Phi_{\text{mask-d}}$. All the components are kept frozen and only the inference is performed. The overall process can be illustrated as follows:

$$\begin{aligned} F_{\text{img}} &= \Phi_{\text{img-e}}(\mathcal{I}), \\ F_{\text{pmt}} &= \Phi_{\text{pmt-e}}(\{P_{\text{box}}, P_{\text{point}}\}), \\ \mathcal{M} &= \Phi_{\text{mask-d}}(F_{\text{img}}, F_{\text{pmt}}), \end{aligned} \tag{5}$$

where $\mathcal{I}$ indicates the original image, $F_{\text{img}}$ represents the image features encoded by $\Phi_{\text{img-e}}$, $F_{\text{pmt}}$ denotes prompt features encoded by $\Phi_{\text{pmt-e}}$, and $\{P_{\text{box}}, P_{\text{point}}\}$ indicates the box and point prompts, respectively. With semantic-explicit prompts, SAM can further refine the target region to cover more complete objects and generate accurate masks.

## 4 EXPERIMENTS

### 4.1 EXPERIMENT SETTINGS

**Datasets.** We evaluate the proposed method on two benchmarks: MS COCO 2017 Lin et al. (2014), Pascal VOC 2007 (Everingham et al., 2015). **MS COCO 2017** is a widely used large-scale dataset that contains 115K images in the *train* set and 5K images in *val* set, with 80 object categories collected in natural scenes. In **Pascal VOC 2007**, there are 2501, 2510, and 4952 images in training, validation, and test sets, respectively, with 20 categories.

**Evaluation Metrics.** We use AP for MS COCO and VOC to measure the performance of detection and segmentation. And we report AP, $AP_{50}$, $AP_{75}$ for MS COCO and $AP_{50}$ for VOC. The mIoU and Correct Localization (CL) are also calculated to directly measure the quality of the pseudo boxes. Specifically, mIoU is calculated by the mean IoU between predicted pseudo boxes and their corresponding ground-truth bounding boxes of all objects in the training set. CL, denoting the correct localization rate, is computed as the ratio of IoU between the prediction and the ground truth

exceeding a certain threshold. We report CL at thresholds of 0.5, 0.7, and 0.9 (termed CL@0.5, CL@0.7, and CL@0.9, respectively) to assess the quality of pseudo boxes.

**Implementation Details.** For SEPG, it is trained with SGD (Robbins & Monro, 1951) optimizer with batch size 16. The learning rate is initialized as 0.02, and decays by a magnitude after 8 and 11 epochs. For SAM (Kirillov et al., 2023), we use the ViT-H version and freeze all the components in SAM. To obtain object detections and segmentations, we choose the classic Faster RCNN (FR) and Mask RCNN (MR), respectively. More implementation details are presented in the Appendix.

Table 1: Comparison of our P2P with other SOTA methods under different forms of supervision on MS COCO 2017 *val* set. Specifically, $\mathcal{F}$, $\mathcal{I}$, and $\mathcal{P}$ indicate full, image-level, and point-level supervision respectively. $\dagger$ denotes fully-supervised refinement.

| Method | Backbone | Sup. | AP | $AP_{50}$ | $AP_{75}$ |
|---|---|---|---|---|---|
| Faster R-CNN (Ren et al., 2015) | R-50 | $\mathcal{F}$ | 37.4 | 58.1 | 40.4 |
| RetinaNet (Lin et al., 2017) | R-50 | $\mathcal{F}$ | 36.5 | 55.4 | 39.1 |
| Reppoint (Yang et al., 2019) | R-50 | $\mathcal{F}$ | 37.0 | 56.7 | 39.7 |
| Sparse R-CNN (Sun et al., 2021) | R-50 | $\mathcal{F}$ | 37.9 | 56.0 | 40.5 |
| DINO (Zhang et al., 2022) | R-50 | $\mathcal{F}$ | 49.0 | 66.4 | 53.3 |
| PCL (Tang et al., 2018) | VGG-16 | $\mathcal{I}$ | 8.5 | 19.4 | - |
| WSOD2 (Zeng et al., 2019) | VGG-16 | $\mathcal{I}$ | 10.8 | 22.7 | - |
| ICMWSD (Ren et al., 2020a) | R-50 | $\mathcal{I}$ | 12.6 | 26.1 | - |
| CASD (Huang et al., 2020) | R-50 | $\mathcal{I}$ | 13.9 | 27.8 | - |
| SPE (Liao et al., 2022) | CaiT | $\mathcal{I}$ | 7.2 | 18.2 | 4.8 |
| WSCL (Seo et al., 2022) | R-50 | $\mathcal{I}$ | 13.8 | 27.8 | 12.1 |
| JLWSOD (Qi, 2023) | R-50 | $\mathcal{I}$ | **14.9** | **29.8** | - |
| UFO$^2$ (Ren et al., 2020b) | R-50 | $\mathcal{P}$ | 13.2 | 28.9 | - |
| P2BNet-FR$^\dagger$ (Chen et al., 2022) | R-50 | $\mathcal{P}$ | 22.1 | 47.3 | - |
| SAM-FR$^\dagger$ (Kirillov et al., 2023) | R-50 | $\mathcal{P}$ | 27.3 | 45.3 | 28.5 |
| Ours-FR$^\dagger$ | R-50 | $\mathcal{P}$ | **31.6** | **53.8** | **32.7** |

Table 2: The performance comparison of fully-supervised ($\mathcal{F}$), image-supervised ($\mathcal{I}$), and point-supervised ($\mathcal{P}$) detectors on Pascal VOC dataset. $^*$ indicates our re-implemented results and $^\dagger$ denotes fully-supervised refinement.

| Method | Set | Backbone | Sup. | $AP_{50}$ |
|---|---|---|---|---|
| Faster R-CNN$^*$ (Ren et al., 2015) | 07 | R-50 | $\mathcal{F}$ | 71.5 |
| WSDDN (Bilen & Vedaldi, 2016) | 07 | R-50 | $\mathcal{I}$ | 39.3 |
| OICR (Tang et al., 2017) | 07 | R-50 | $\mathcal{I}$ | 42.0 |
| PCL (Tang et al., 2018) | 07 | R-50 | $\mathcal{I}$ | 45.8 |
| MELM (Wan et al., 2018) | 07 | R-50 | $\mathcal{I}$ | 47.1 |
| W2F$^\dagger$ (Zhang et al., 2018) | 07 | R-50 | $\mathcal{I}$ | 52.4 |
| CASD (Huang et al., 2020) | 07 | R-50 | $\mathcal{I}$ | 56.8 |
| SPE (Liao et al., 2022) | 0712 | CaiT | $\mathcal{I}$ | 51.0 |
| P2BNet-FR$^{*\dagger}$ (Chen et al., 2022) | 07 | R-50 | $\mathcal{P}$ | 48.3 |
| SAM-FR$^\dagger$ (Kirillov et al., 2023) | 07 | R-50 | $\mathcal{P}$ | 47.9 |
| Ours-FR$^\dagger$ | 07 | R-50 | $\mathcal{P}$ | **61.9** |

Table 3: The segmentation performance of fully supervised $\mathcal{F}$, box-supervised $\mathcal{B}$, image-supervised $\mathcal{I}$, and point-supervised $\mathcal{P}$ methods on MS COCO 2017 *val* set. $^\dagger$ denotes fully-supervised refinement.

| Method | Backbone | Sup. | AP | $AP_{50}$ | $AP_{75}$ |
|---|---|---|---|---|---|
| Mask R-CNN (He et al., 2017) | R-50 | $\mathcal{F}$ | 35.4 | 56.4 | 37.9 |
| Mask R-CNN (He et al., 2017) | ViT | $\mathcal{F}$ | 38.8 | 61.2 | 41.3 |
| Mask2Former (Cheng et al., 2022) | Swin-S | $\mathcal{F}$ | 46.1 | 69.4 | 49.8 |
| BESTIE (Kim et al., 2022) | HRNet48 | $\mathcal{I}$ | 14.3 | 28.0 | 13.2 |
| BoxInst (Tian et al., 2021) | R-50 | $\mathcal{B}$ | 32.1 | 55.1 | 32.4 |
| SAM-MR$^\dagger$ Kirillov et al. (2023) | R-50 | $\mathcal{B}$ | 31.1 | 53.1 | 47.1 |
| WISE-Net (Laradji et al., 2020) | R-50 | $\mathcal{P}$ | 7.8 | 18.2 | 8.8 |
| BESTIE$^\dagger$ (Kim et al., 2022) | HRNet48 | $\mathcal{P}$ | 17.7 | 34.0 | 16.4 |
| AttnShift$^\dagger$ (Liao et al., 2023) | ViT | $\mathcal{P}$ | 21.2 | 42.0 | 19.4 |
| SAM-MR$^\dagger$ (Kirillov et al., 2023) | R-50 | $\mathcal{P}$ | 24.3 | 43.8 | 24.3 |
| Ours-MR$^\dagger$ | R-50 | $\mathcal{P}$ | **26.4** | **48.6** | **26.2** |

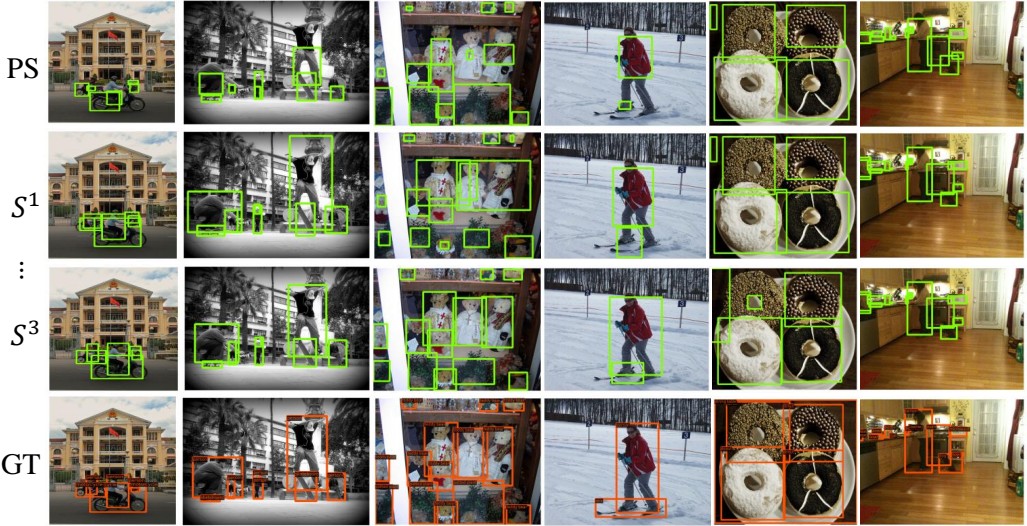

Figure 3: Visualization of the pseudo bounding boxes from different iterations of P2P and Ground Truths (GT). PS denotes the proposal seeds. (Best viewed in color.)

## 4.2 Performance and Comparison

Our method serves as a pseudo-boxes and -masks generator. Employing basic FR and MR with pseudo-boxes and -masks in a fully supervised manner, we report the detection and segmentation performance of our method. Furthermore, we conduct comparative analyses by benchmarking our method against SOTA fully supervised methods to reflect their performance upper bound and with various forms of weakly supervised methods to show the effectiveness of point-supervised methods.

**Detection Performance.** We conduct comparisons between our method and fully, image-, and point-supervised detection methods using COCO and VOC datasets, as presented in Tab. 1 and 2. On the COCO dataset, our method outperforms the SOTA P2BNet (Chen et al., 2022) by 9.5% (31.6% vs 22.1%) and 6.5% (53.8% vs 47.3%) in terms of $AP$ and $AP_{50}$, respectively, and achieves 84% of the fully supervised FR's performance. On the VOC dataset, our method surpasses previous SOTA by 5.1% and achieves 86% of the performance of fully supervised FR. From Tab. 1, we find that the image-supervised methods perform poorly on the challenging COCO dataset, achieving only about 36% of the fully-supervised baseline. This is notably lower than the performance of the point-supervised method, highlighting the favorable trade-off between labeling burden and performance offered by the point-supervised approach. Additionally, using SAM directly as a pseudo-boxes generator with FR (SAM-FR) yields comparable performance on both datasets, with 45.3% and 47.9% $AP_{50}$ on COCO and VOC, respectively.

**Segmentation Performance.** Tab. 3 gives the performance of the segmentation methods on the COCO dataset with different forms of supervision. The P2P method reports a significant performance improvement of 5.2% AP over the AttnShift (Liao et al., 2023) approach and outperforms the image-supervised BESTIE (Kim et al., 2022) by a large margin. Additionally, it also outperforms SAM-MR (where SAM is used as a pseudo-masks generator) by 2.1% AP and 4.8% $AP_{50}$. Furthermore, our method with only point supervision achieves about 75% and 82% of the performance exhibited by fully supervised and box-supervised methods with ResNet-50 backbone, respectively.

## 4.3 Ablation Study

We conduct analyses of the impact of key components of our method on the COCO dataset.

**Effect of each component in P2P.** The ablation study of each component in our approach is shown in Tab. 4. P2BNet is used as the baseline, and the key components include (i) **SS**: When seeds-based sampling policy with the refinement head in the baseline model is applied, we observe a slight decrease in performance compared to the baseline. This suggests that the utilization of seeds

Table 4: Effect of each component in P2P. **SS** stands for seeds-based sampling.

| Method | SS | SEPG | PGSR | Iter | mIoU | AP | AP$_{50}$ | AP$_{75}$ |
|---|---|---|---|---|---|---|---|---|
| Baseline | - | - | - | - | 57.5 | 22.1 | 47.3 | - |
| Ours | ✓ | | | | 56.6 | 21.1 | 44.9 | 17.1 |
| | ✓ | ✓ | | | 60.3 | 23.1 | 49.6 | 18.1 |
| | ✓ | ✓ | ✓ | | 68.1 | 30.4 | 52.7 | 31.2 |
| | ✓ | ✓ | ✓ | ✓ | 69.7 | 31.5 | 53.1 | 32.9 |

Table 5: Effect of different techniques in SEPG stage. SEPG-base stands for seeds-based sampling, SEPG-G stands for group selection, and SEPG-G-SC stands for semantic confidence.

| Method | mIoU | AP | AP$_{50}$ | AP$_{75}$ |
|---|---|---|---|---|
| SEPG-base | 56.6 | 21.1 | 44.9 | 17.1 |
| SEPG-G | 55.2 | 21.5 | 46.6 | 16.9 |
| SEPG-G-SC | 60.3 | 23.1 | 49.6 | 18.1 |

Table 6: Effect of different threshold $\tau$.

| $\tau$ | mIoU | CL@0.5 | CL@0.7 | CL@0.9 |
|---|---|---|---|---|
| 0.6 | 60.00 | 74.03 | 41.05 | 2.10 |
| 0.7 | **60.28** | **74.47** | 41.87 | 2.13 |
| 0.8 | 60.20 | 74.33 | **41.96** | **2.14** |
| 0.9 | 60.03 | 74.07 | 41.54 | 2.11 |

Table 7: Effect of different iteration $T$.

| $T$ | mIoU | CL@0.5 | CL@0.7 | CL@0.9 |
|---|---|---|---|---|
| 1 | 68.07 | 76.77 | 60.27 | 23.10 |
| 2 | 69.43 | 78.74 | 62.13 | 24.09 |
| 3 | **69.70** | **79.05** | **62.40** | 24.07 |
| 4 | 69.61 | 78.00 | 61.91 | **24.35** |

does not directly lead to a performance improvement. (ii) **SEPG**: The group sampling technique reduces the solution space, and more accurate pseudo-labels are obtained with the two semantic confidence-guided refiners. This contribute to an improvement in mIoU by more than 3 points. (iii) **PGSR**: SAM leverages semantic-explicit prompts for spatial refinement of object regions, leading to a substantial enhancement in the quality of pseudo-labels, with an improvement of about 8 points. (iv) Employing an **Iter**ative strategy where the two models mutually enhance each other, we observe a notable 12% improvement in performance compared to the baseline model.

**Effect of different techniques in SEPG.** We validate different techniques in SEPG and the results are presented in Tab. 5. Initially, the baseline model is designed by adopting the seeds-based sampling strategy followed by cascaded MIL refinement (same to **SS** in Tab. 4), referred to "SEPG-base", achieving 21.3 AP and 44.9 AP$_{50}$. Then, we adopt group refinement, *i.e.*, selecting groups first and then individual proposals. When using the MIL refinement head, referred to as "SEPG-G", AP reaches 21.5, as shown in the second row of Tab. 5. Further improvement is achieved with our designed semantic confidence refinement head, denoted as "SEPG-G-SC", resulting in an mIoU of 60.3 and an AP of 23.1, as demonstrated in the fourth row of Table 5.

**Effect of different thresholds.** We analyze the impact of varied thresholds $\tau$ that are used to update semantic prototype, and the results are detailed in Tab. 6. $\tau$ plays a crucial role in controlling the quality of the semantic prototypes. When $\tau$ is set to a lower value, it involves a larger number of embedding features, potentially resulting in lower-quality updates to the semantic prototype. Conversely, when $\tau$ is set to a higher value, only a limited number of embedding features contribute to updating the semantic prototype, which may not effectively represent global features. Observations from our study indicate that as $\tau$ increases, the quality of the semantic prototypes improves, thereby bringing better performance. The peak performance is achieved when $\tau = 0.7$.

**Effect of Iterative Training.** We examine the impact of training iterations $T$ on prediction quality. Tab. 7 reports the performance of P2P with different iteration numbers. We observe that the value of mIoU is consistently improved as $T$ increases, reaching a peak when $T = 3$, followed by a stabilization around the peak. Notably, the highest performance surpasses that of $T = 1$ by nearly 2% across all metrics, showing the effectiveness of iterative training in elevating the quality of predictions. Some representative visualizations of different iterations are presented in Fig. 3.

## 5 CONCLUSION

In this paper, we introduce a point-supervised object detection and segmentation framework, called P2P, which transforms weak point-level annotations into explicit visual prompts, and guides the foundation model to produce the desired output by improving the semantic representation of the prompts. P2P performs as an iterative procedure that includes two stages: SEPG and PGSR. Taking the proposal seeds as input, SEPG improves the semantic confidence of the proposals through group sampling and semantic prototypes, and produces semantic explicit boxes as prompts. With carefully designed prompts, the PGSR stage leverages SAM to output refined masks, which in turn are transformed into new proposal seeds. Finally, accurate pseudo masks and boxes are obtained in the iteration of SEPG and PGSR. Extensive experiments validate the effectiveness of our method.

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

# A APPENDIX

## A.1 MORE DETAILS

**Details of Group Sampling.** For the first phase, we introduce a seeds-based group sampling strategy. With the point annotation $p = (p_x, p_y)$, the corresponding proposal seed $b = (b_x, b_y, b_w, b_h)$ is generated by SAM. We first adjust the scale of the proposal seed with $s$ to get $n$ bases, then the proposal group is generated by sampling proposals with strong spatial correlation around each base. Finally, $n$ proposal groups construct the proposal bag $\mathcal{B}_1$. We empirically set $r = \{1/3, 1/2, 1, 2, 3\}$. For the second phase, we adopt a similar sampling strategy like Chen et al. (2022), *i.e.*, adjusting the width and height with $s_w$ and $s_h$ of the proposals in the $B^*$, and jitter its position with $o_x, o_y$. We set $s_w \in \{0.7, 0.8, 1, 1.2, 1.3\}$ and $s_h \in \{0.7, 0.8, 1, 1.2, 1.3\}$.

**Details of P2P Network.** As shown in Fig. 2, the designed SEPG adopts the coarse-to-fine policy in a cascaded structure, which consists of two similar refiners that share some of the weights. For both refiners, region features of the proposals are extracted by backbone network (with FPN) and two FC+ReLU blocks. And then the proposal scores and the embedding features could be obtained by classification head $\Phi_{cls}$ and projection head $\Phi_{proj}$, respectively. The parameters of the two refiners are shared in both the backbone network and the two FCs. Specifically, in *Prompt Refiner I*, taking the proposal bag $\mathcal{B}_1$ as input, region features $F \in \mathbf{R}^{N \times D}$ are extracted, which will be firstly processed by two fully connected (fc) layers with ReLU function and a Softmax head to obtain the proposal scores $O \in \mathbf{R}^{N \times C}$, where $N$ denotes the number of proposals in $\mathcal{B}_1$, $D$ denotes the feature dimension, and $C$ denotes the number of classes. Meanwhile, $F$ will also be mapped to the embedding space on top of a two-layer non-linear projection head to generate embedding features $Z \in \mathcal{R}^{N \times D'}$, where $D'$ denotes embedding dimension, default is 128. After that, semantic confidence $\mathcal{P}_{sem}$ and instance confidence $\mathcal{P}_{ins}$ can be calculated by proposal scores $O$ (Eq. 2) and embedding similarity (Eq. 1). The proposal group $B^*$ with the highest semantic confidence is selected and taken as the input of the second refiner.

The *Prompt Refiner II* performs further refining with a similar structure as the first phase. The proposal bag $\mathcal{B}_2$ is first built by a "box jittering" strategy to augment the proposals in $B^*$. we only employ the general classification head with sigmoid function and adopted the focal loss (termed $\mathcal{L}_2$) (Lin et al., 2017) for further refinement, as

$$\mathcal{L}_2 = -\sum_{c=1}^{C} y_c(1-S)^\gamma \log S + (1-y_c)(S)^\gamma \log(1-S), \tag{6}$$

where $S$ denotes the score of the proposal bag, which is the average of the scores of all the proposals in $\mathcal{B}_2$. The proposals with the top-$k$ highest scores are weighted to obtain the final refined box $b^*$, where $k$ is set to 4 by default. The overall loss function is the summation of the losses from these two phases, *i.e.*, $\mathcal{L} = \mathcal{L}_1 + \mathcal{L}_2$.

**Details of Implementation.** We implement our method with 4 NVIDIA RTX 4090 GPUs. For SEPG, it is trained with SGD (Robbins & Monro, 1951) optimizer with batch size 4 per GPU. The learning rate is initialized as 0.02, and decays by a magnitude after 8 and 11 epochs. We adopt ResNet-50 (He et al., 2016) pre-trained on ImageNet-1K (Russakovsky et al., 2015) as the backbone network. We use multi-scale (480, 576, 688, 864, 1000, 1200) as the short side to resize the image during training and single-scale (1200) during inference. For a fair comparison, we adopt the quasi-center method proposed by Chen et al. (2022) by default without modifying its hyperparameters to generate point annotations for the dataset. For SAM (Kirillov et al., 2023), we use the ViT-H version and freeze all the components in SAM. To obtain object detections and segmentations, we choose the classic Faster RCNN (Ren et al., 2015) and Mask RCNN (He et al., 2017) (the backbone of both is ResNet-50 with FPN), respectively. Furthermore, we also employ more advanced detection and segmentation networks, such as DINO (Zhang et al., 2022) and Mask2Former (Cheng et al., 2022), to validate the effectiveness of our approach. For all detection and segmentation experiments, we use the default setting in MMDetection (Chen et al., 2019).

## A.2 MEMORY AND TIME COST.

Tab. 8 illustrates the memory and time consumption of SEPG and PGSR. We tested on a $4 \times$ RTX 4090 platform and used COCO dataset. All methods are tested using a batch size of 4 on a single

GPU. SEPG leads to a bit higher memory and time cost in training than P2BNet. The parameters are fixed during PGSR stage, and the process takes approximately 2.4 hours. Tab. 9 displays the total training time consumption under different number of iterations. When the number of iteration is 1 ($T = 1$), we train SEPG for 12 epochs. For $T > 1$, the number of epochs is set to 6 to reduce the training time cost.

Table 8: Memory, training time (t/Epoch) and inference time (Inf. time) cost.

| Method | Model | Epoch | Mem. (M) | t/Epoch (h) | Inf. time (s) |
|--------|-------|-------|----------|-------------|---------------|
| P2BNet | R-50 | 12 | 4074 | $\sim 1.08$ | 0.078 |
| SEPG | R-50 | 12 or 6 | 4598 | $\sim 1.25$ | 0.090 |
| PGSR | ViT-H | - | 7696 | $\sim 2.40$ | 0.292 |

Table 9: Effect of different iteration $T$.

| $T$ | Epochs/iter | Total time (h) |
|-----|-------------|----------------|
| 1 | 12 | $\sim$17 |
| 2 | 6 | $\sim$20 |
| 3 | 6 | $\sim$30 |
| 4 | 6 | $\sim$40 |

### A.3 SUPPLEMENTARY EXPERIMENTS

**Better Detectors and Segmentation Networks.** Following the default setting, we retrain DINO and Mask2Former by 12 and 50 epoch, respectively, supervised by the pseudo labels from teh 3rd iteration of P2P. The backbone of P2P is ResNet-50. As shown in Tab. 10, 1) When DINO (Zhang et al., 2022) (both R-50 and Swin-L are used as backbone) is adopted as the detector, our method achieves a 38.2 AP (R-50) and 45.1 AP (Swin-l) on COCO dataset, approximately 80% of the fully supervised performance, and a 57.2 AP50 (R-50) and 66.1 $AP_{50}$ (Swin-l), about 87% of the fully supervised performance. 2) When the Mask2Former (Cheng et al., 2022) is employed as the segmentation network, achieving a 34.9 AP and 58.9 $AP_{50}$. In summary, when utilizing more advanced detector or segmentation network, our method still closely approaches the performance of fully supervised methods.

**Detailed Comparison with SAM.** As shown in Tab. 11, we conducted a more detailed comparison with SAM. For mIoU and CorLoc, our approach achieved significant improvements, increasing by 11.45 (mIoU) and 18.73 (for CL@0.5) points, respectively. This indicates that our method effectively enhances the quality of pseudo labels. Additionally, our approach also demonstrates a notable improvement in $AP_{50}$, with a performance increase of 8.5 on COCO dataset.

### A.4 MORE VISUALIZATIONS

Fig. 4 shows the evolution of predicted results in one iteration. Starting with semantic-agnostic proposal seeds, SEPG progressively identifies the semantic regions. PGSR then takes over to further refine the target region, accurately locating the complete object.

The segmentation results of P2P on the COCO dataset are given in Fig. 5. The visualization of the pseudo bounding boxes of P2P on VOC set is presented in Fig. 6.

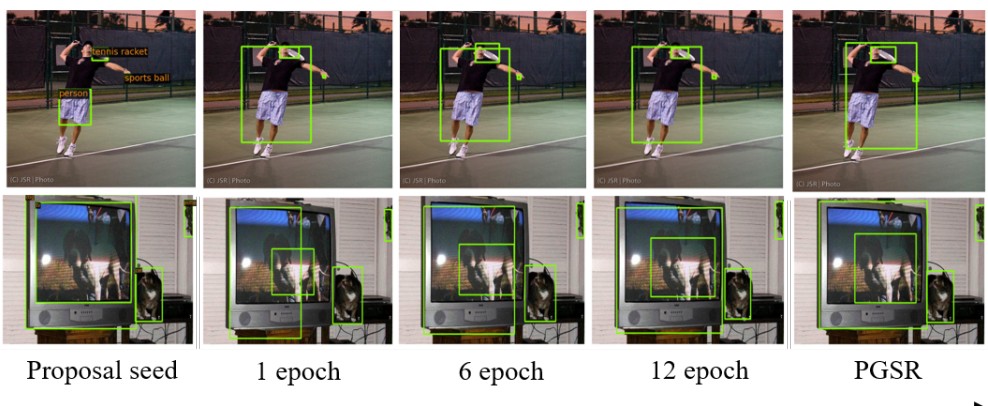

Figure 4: Evolution of predicted results in one iteration. The images are from the COCO 2017 *val* set. (Best viewed in color.)

Table 10: Comparison with more advanced detectors and sementation networks.

| Method | Backbone | Retrained epoch | $AP$ | $AP_{50}$ | $AP_{75}$ |
|---|---|---|---|---|---|
| DINO (Zhang et al., 2022) | R-50 | 12 | 49.0 | 66.4 | 55.3 |
| DINO (Zhang et al., 2022) | Swin-l | 12 | 57.2 | 75.7 | 62.7 |
| Ours-DINO | R-50 | 12 | 38.2 | 57.2 | 40.9 |
| Ours-DINO | Swin-l | 12 | 45.1 | 66.1 | 48.9 |
| Mask2Former (Cheng et al., 2022) | Swin-s | 50 | 46.1 | 69.4 | 49.8 |
| Ours-MF | Swin-s | 50 | 34.9 | 58.9 | 37.6 |

Table 11: Detailed comparison with SAM.

| Method | mIoU | CL@0.5 | CL@0.7 | CL@0.9 | AP | $AP_{50}$ | $AP_{75}$ |
|---|---|---|---|---|---|---|---|
| SAM (Kirillov et al., 2023) | 58.25 | 60.32 | 47.32 | 21.27 | 27.3 | 45.3 | 28.5 |
| Ours | 69.70 | 79.05 | 62.40 | 24.07 | 31.6 | 53.8 | 32.7 |

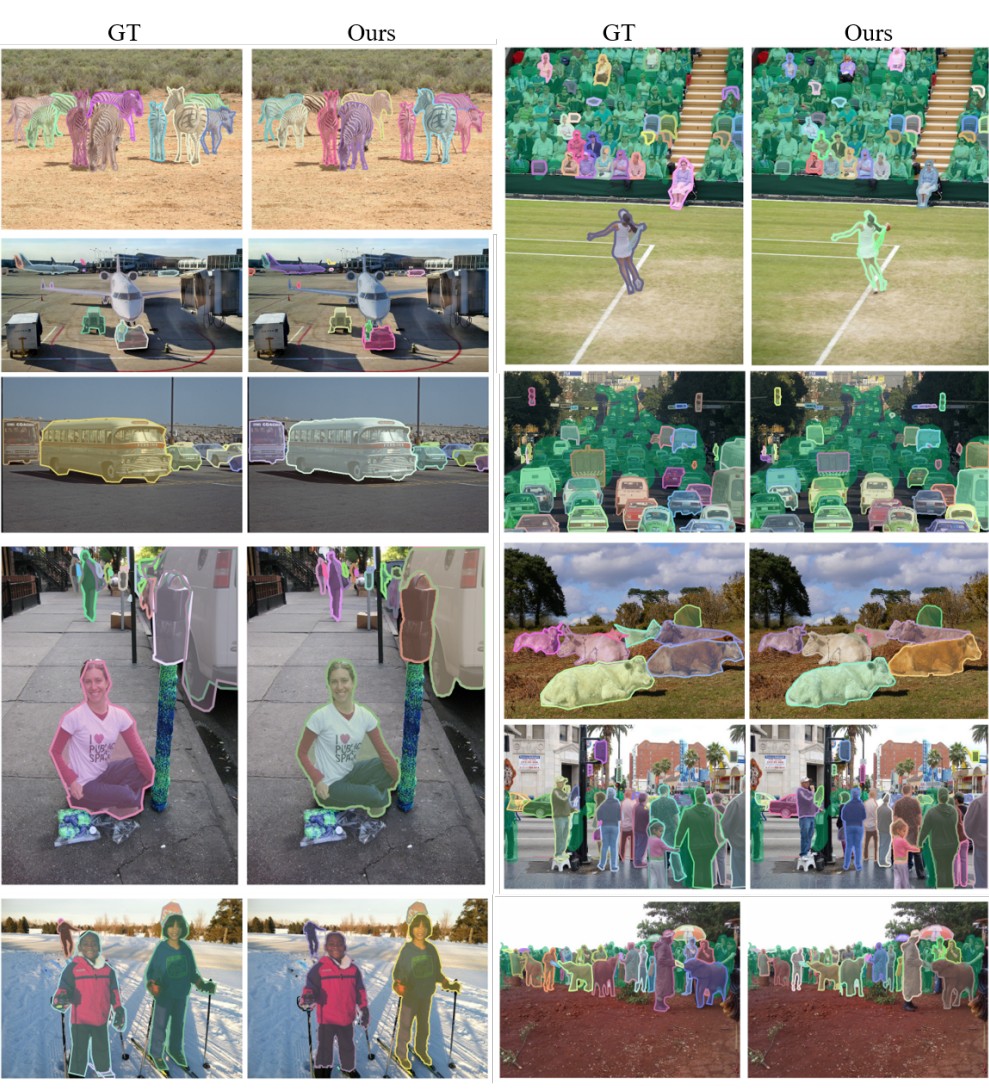

Figure 5: The segmentation visualization of P2P. The images are from COCO 2017 *val* set. (Best viewed in color.)

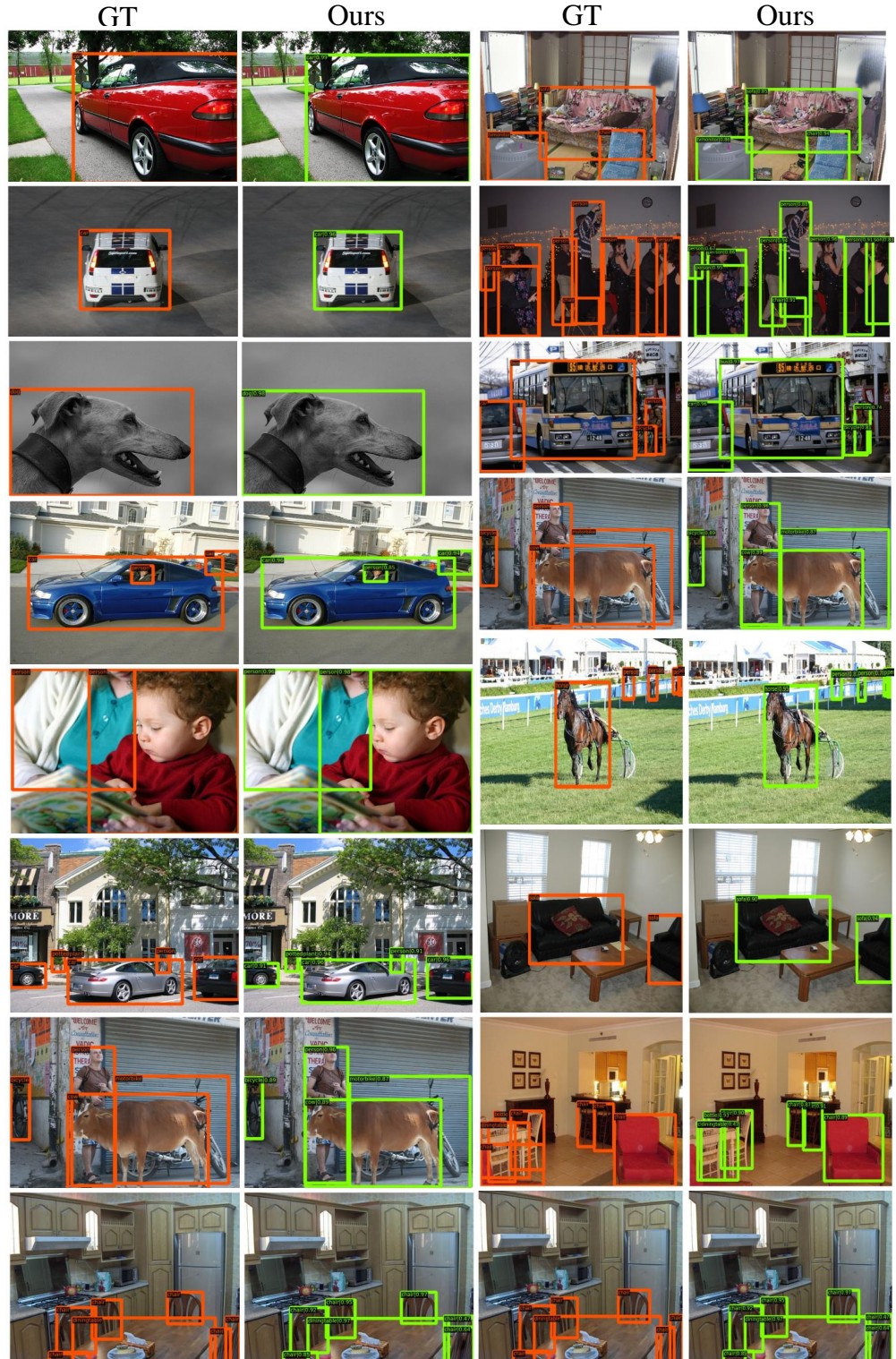

Figure 6: Visualization of the pseudo bounding boxes of P2P and Ground Truths. The images are from VOC 2007 test set. (Best viewed in color.)

