# OpenReview forum: "P2P: Transforming from Point Supervision to Explicit Visual Prompt for Object Detection and Segmentation"
_ICLR.cc/2024/Conference — Submitted to ICLR 2024_

### Official Review · Reviewer_6Xmh · 2023-10-24

**Soundness:** 3 good
**Presentation:** 3 good
**Contribution:** 3 good
**Rating:** 5
**Confidence:** 4

**Summary:**

This paper concentrates on the task of point supervision and introduces a novel approach named  P2P that transforms point supervision into accurate pseudo-labels. The method harnesses the potential of the visual foundation model, SAM, and introduces an iterative framework for the generation of these pseudo-labels. This framework comprises two stages: an SEPG stage, which translates the point annotations into visual prompts, and a PGSR stage for converting these visual prompts into pseudo masks and bounding boxes. Experimental results show that the proposed approach achieves SOTA performance on COCO and PaSCAL datasets.

**Strengths:**

- This paper introduces a novel method to generate pseudo-labels from point supervision that leverages the potential with a visual foundation model.
- The paper is well-written and experiments demonstrate superior qualitative results  compared to baselines.

**Weaknesses:**

- The performance of the model largely depends on the capability of SAM itself. From the quantitative results, the proposed approach does not have a significant improvement compared with directly using SAM.
- The method seems to be computationally demanding since the training process is conducted with 4*RTX 4090 GPUs.

**Questions:**

Could the author provide a comparison of training time and memory cost compared with other methods?

---

> ### Author Response · Authors · 2023-11-20
> **Response to reviewer 6Xmh**
>
> Thanks for your constructive comments. We hope our responses can address your concerns. Further comments are welcomed.
>
> ------
>
> Q1: *The performance of the model largely depends on the capability of SAM itself. From the quantitative results, the proposed approach does not have a significant improvement compared with directly using SAM.*
>
> A1: Thanks for your comment. The emergence of large models has brought new insights and performance improvements to many downstream tasks. We think that the trend of applying foundation models to solve problems is irreversible.
>
> For point-supervised detection and segmentation, the absence of size information makes it challenging. SAM, due to its lack of semantic understanding, also struggles to fully address this issue. In our method, P2P is the first attempt to iteratively enhance the quality of prompts, guiding SAM to generate accurate masks and achieving further improvement upon SAM.
>
> we conducted a more detailed comparison with SAM. We found that, for mIoU and CorLoc, our approach improves mIoU and CL@0.5 by 11.4 and 18.37, respectively. Additionally, our approach also demonstrates a notable improvement in $AP_{50}$, with a performance increase of 8.5 on COCO dataset.
>
>
>
> | Method | mIoU  | CL@0.5 | CL@0.7 | CL@0.9 |  AP  | AP50 | AP75 |
> | :----: | :---: | :----: | :----: | :----: | :--: | :--: | :--: |
> | P2BNet | 57.5  |   -    |   -    |   -    | 22.1 | 47.3 |  -   |
> |  SAM   | 58.25 | 60.32  | 47.32  | 21.27  | 27.3 | 45.3 | 28.5 |
> |  Ours  | 69.70 | 79.05  | 62.40  | 24.07  | 31.6 | 53.8 | 32.7 |
>
>
>
> Q2: *The method seems to be computationally demanding since the training process is conducted with 4\*RTX 4090 GPUs. Could the author provide a comparison of training time and memory cost compared with other methods?*
>
> A2: As suggested, we have supplemented detailed comparisons of memory, training time, and inference time in the appendix of revised version. This is also explained in the *unified response* part.
>
> P2P consumes a bit higher time and memory during the training phase than P2BNet, but it greatly enhances the quality of pseudo-labels, which is crucial for subsequent detection and segmentation tasks. With high-quality pseudo-labels, one can freely choose any task-specific subnetwork for optimization, achieving commendable performance. Therefore, compared to the substantial improvement in performance, the time consumption during training is not an issue.

---

> ### Author Response · Authors · 2023-11-22
> **Response to reviewer 6Xmh (2st)**
>
> Dear Reviewer 6Xmh,
>
> Thank you very much for reviewing our work. Your comments and feedback will be invaluable in helping us improve the final version of our paper.
>
> We think we have responded to the reviewer's questions and concerns, but if there is anything that is missing, please let us know and we will be happy to add it.
>
> Looking forward to your feedback and discussion. Thanks again!
>
> Best,
>
> Paper 1568 Authors.

---

### Official Review · Reviewer_j4vZ · 2023-10-28

**Soundness:** 3 good
**Presentation:** 3 good
**Contribution:** 3 good
**Rating:** 6
**Confidence:** 4

**Summary:**

This paper proposes a novel iterative learning framework, Point to Prompt (P2P), for point-supervised object detection and segmentation. The P2P is formulated as an iterative refinement process of two stages: Semantic Explicit Prompt Generation (SEPG) and Prompt Guided Spatial Refinement (PGSR). Experiments on multiple datasets are performed.

**Strengths:**

- The proposed method has a superior performance on multiple datasets.
- The motivation is resonable and easy to understand.

**Weaknesses:**

- The authors claim that the existing methods aim to release the annotation burden while still achieving decent performance. However, the proposed method seems still far behind fully-supervised method.
- It is better to give an inference time comparison, which can help readers better understand the propose method.
- SAM has a good ability to generate the mask of objects based on point. Thus, it may give a good performance using SAM and Cascade structure, which is similar to PGSR.

**Questions:**

please see weakness

---

> ### Author Response · Authors · 2023-11-20
> **Response to reviewer j4vZ**
>
> Thanks for your constructive comments. We hope our responses can address your concerns. Further comments are welcomed.
>
> -----
>
> Q1: *The authors claim that the existing methods aim to release the annotation burden while still achieving decent performance. However, the proposed method seems still far behind fully-supervised method.*
>
> A1:
>
> 1. **point-supervised tasks is challenging:** Due to the absence of target size information, accurate detection and segmentation are challenging under point supervision alone. In theory, the performance of fully supervised methods serves as an upper bound for the performance of all weakly supervised methods. As shown in the following tables, under equivalent conditions (i.e., point supervision using the same fully supervised method fine-tuned), previous state-of-the-art point-supervised methods achieved only around **60% (for detection) or 50% (for segmentation)** of the fully supervised performance.
> 2. **We further narrows the performance gap:** With only point annotations, our method can achieve performance levels of around **80%** of a fully supervised detector, significantly reducing the required annotation effort.
>
>
>
> |    Method    | Sup. |  AP  | AP50 | AP75 |
> | :----------: | :--: | :--: | :--: | :--: |
> | Faster  RCNN |  F   | 37.4 | 58.1 | 40.4 |
> |     DINO [1]     |  F   | 57.2 | 75.7 | 62.7 |
> |  P2BNet-FR   |  P   | 22.1 | 47.3 |  -   |
> |   Ours-FR    |  P   | 31.6 | 53.8 | 32.7 |
> |  Ours-DINO   |  P   | 45.1 | 66.1 | 48.9 |
>
> |    Method     | Sup. |  AP  | AP50 | AP75 |
> | :-----------: | :--: | :--: | :--: | :--: |
> |   Mask RCNN   |  F   | 35.4 | 56.4 | 37.9 |
> |  Mask2Former [2]  |  F   | 46.1 | 69.4 | 49.8 |
> | AttenShift-MR |  P   | 21.2 | 42.0 | 19.4 |
> |    Ours-MR    |  P   | 26.4 | 48.6 | 26.2 |
> |    Ours-MF    |  P   | 34.9 | 58.9 | 36.1 |
>
> *Note: FR indicates Faster RCNN, MR indicates Mask RCNN, and MF indicates Mask2Former*
>
> ------
>
> *[1] Zhang, Hao, et al. "DINO: DETR with Improved DeNoising Anchor Boxes for End-to-End Object Detection." The Eleventh International Conference on Learning Representations. 2022.*
>
> *[2] Cheng, Bowen, et al. "Masked-attention mask transformer for universal image segmentation." Proceedings of the IEEE/CVF conference on computer vision and pattern recognition. 2022.*
>
> ------
> Q2: *It is better to give an inference time comparison, which can help readers better understand the propose method.*
>
> A2: As suggested, we have supplemented detailed comparisons of memory, training time, and inference time in the appendix of revised version. This is also explained in the *unified response* part.
>
> For P2P, the inference speed is a bit lower than P2BNet. However, we think that the time consumption of point-supervised methods itself may not be a primary limitation in practical applications. As illustrated  in the section 3.1, the point supervision task involves training a regressor to predict pseudo-labels. One can choose an appropriate subnetwork (*e.g.*, a detector ) for specific tasks or applications and opimized it in a fully supervised manner with the predicted pseudo labels, (*e.g.*, we can use more powerful detectors, such as DINO, to ensure performance, or choose faster detectors like YOLO to guarantee real-time capabilities.).
>
>
>
> Q3: *SAM has a good ability to generate the mask of objects based on point. Thus, it may give a good performance using SAM and Cascade structure, which is similar to PGSR.*
>
> A3: Thank you for your comment.
>
> **Results**: Following your advice, we devised the cascaded-SAM. However, experimental results (in the following table) indicate that it **did not yield satisfactory performance**.
>
> **Analysis**: We attribute SAM's performance to the quality of prompts. A critical issue with point prompts is semantic ambiguity. SAM cannot ascertain whether the semantic of a point is local or global (e.g., a car or a car window). Under point prompts, numerous local masks are generated (e.g., clothing on a person, the glass of a car), which becomes more pronounced in complex scenes, such as the COCO dataset. Even when using the bounding box of local masks as prompts and reapplying SAM, the resulting masks remain local and fail to enhance performance effectively. In contrast, **PGSR progressively regresses proposal seeds to the correct semantic regions based on given semantic labels**, (e.g., transitioning from clothing to the entire person), which used as a prompt can better guide SAM in generating accurate masks.
>
> |   Method    | mIoU  | CL@0.5 | CL@0.7 | CL@0.9 |  AP  | AP50 | AP75 |
> | :---------: | :---: | :----: | :----: | :----: | :--: | :--: | :--: |
> |     SAM     | 58.25 | 60.32  | 47.32  | 21.27  | 24.3 | 43.8 | 24.3 |
> | Cascade SAM | 58.56 | 60.64  | 47.76  | 22.05  | 24.7 | 44.3 | 24.8 |
> |    Ours     | 69.70 | 79.05  | 62.40  | 24.07  | 26.4 | 48.6 | 26.2 |

---

> ### Author Response · Authors · 2023-11-22
> **Response to reviewer j4vZ (2st)**
>
> Dear Reviewer j4vZ,
>
> Thank you very much for reviewing our work. Your comments and feedback will be invaluable in helping us improve the final version of our paper.
>
> We think we have responded to the reviewer's questions and concerns, but if there is anything that is missing, please let us know and we will be happy to add it.
>
> Looking forward to your feedback and discussion. Thanks again!
>
> Best,
>
> Paper 1568 Authors.

---

### Official Review · Reviewer_iqSb · 2023-10-29

**Soundness:** 3 good
**Presentation:** 3 good
**Contribution:** 3 good
**Rating:** 6
**Confidence:** 3

**Summary:**

This paper innovatively introduces the Point to Prompt task, transforming point-label inputs into visual prompt learning, and leveraging a foundation model (SAM). It utilizes an iterative refinement process to obtain high-quality prompt to complete object detection and semantic segmentation tasks.

**Strengths:**

- This method simultaneously accomplishes segmentation and horizontal object detection tasks through point labels, demonstrating a certain level of versatility.
- This paper employs Exponential Moving Average (EMA) to update the prototypes of the corresponding categories. This approach is dynamic and remains unaffected by the inherent instability of Multi Instance Learning.
- Novelty: this paper introduces a novel approach by incorporating visual foundation model into a point-supervised task while serving two downstream tasks simultaneously. Furthermore, the proposed prototype representation updated by Exponential Moving Average (EMA) within the Multiple Instance Learning (MIL) framework can enhance the quality of the selected proposals.
- Writing: The paper exhibits a generally smooth logical flow and suitable symbol usage.

**Weaknesses:**

- Novelty: Some parts of the proposed method lack novelty, particularly in the proposed Prompt Refiner II, phrases like "similar to previous works" and "follow the common practice" resemble the refinement stage of P2BNet [1].
- Writing: Certain parts of the language are not as concise and straightforward as desired. Additionally, there is a slight delay in providing specific explanations for some concepts, such as "proposal seeds" and "selected high-quality embedding features,". Furthermore, Section 3.4 should be repositioned closer to the beginning of the paper to provide readers with a clearer understanding of the overall method's workflow.

[1] Chen, Pengfei, et al. "Point-to-box network for accurate object detection via single point supervision." European Conference on Computer Vision. Cham: Springer Nature Switzerland, 2022.

**Questions:**

- In Section 3.1, the paper mentions that “the core of this task lies in designing an accurate point-to-box regressor”, and introduce the P2P framework (point-to-visual prompt), but it is inconsistent between “point-to-box” and “point-to-visual prompt”, it is suggested to add the description of process of visual prompt-to-box/mask after.
- What’s the meaning of the orange arrow in the lower part of the Figure 2? (i.e., the arrow on the right of ‘Refine box b*’).
- The concept ‘proposal seeds’ is proposed in Section 3.1, but its specific generation way (i.e., outer rectangle of the mask generated by SAM) is given until Section 3.4, it is suggested to explain its specific concept early on to avoid confusion.
- What’s the concrete content of ‘sharing some of the weights’ about the two refiners mentioned in Section 3.2?
- What are the key differences in the workflow of the proposed Refiner2 compared to the refining stage in previous works [1]?
- How about the computational complexity of the proposed method compared to other methods?
- In Section 3.4, the actual input of the subsequent P2P refiner is the initial proposal seed box (i.e., outer rectangle of the mask generated by SAM). If possible, it is recommended to use the box (e.g., the fully supervised bounding box label) as the prompt of SAM to conduct a more comprehensive experimental comparison in segmentation, rather than just using the point as the prompt of SAM for comparison.

[1] Chen, Pengfei, et al. "Point-to-box network for accurate object detection via single point supervision." European Conference on Computer Vision. Cham: Springer Nature Switzerland, 2022.

---

> ### Author Response · Authors · 2023-11-20
> **Response to reviewer iqSb**
>
> # Response to reviewer iqSb
>
> Thanks for your constructive comments. We hope our responses can address your concerns. Further comments are welcomed.
>
> -----
>
> Q1: *In Section 3.1, the paper mentions that “the core of this task lies in designing an accurate point-to-box regressor”, and introduce the P2P framework (point-to-visual prompt), but it is inconsistent between “point-to-box” and “point-to-visual prompt”, it is suggested to add the description of process of visual prompt-to-box/mask after*.
>
> A1: Thanks for your detailed comment. In P2P, the two stages, SEPG and PGSR, respectively serve the roles of ‘point to prompt’ and ‘prompt to pseudo-mask’. We have added this description in Section 3.1 to make the logic more complete.
>
>
>
> Q2: *What’s the meaning of the orange arrow in the lower part of the Figure 2? (i.e., the arrow on the right of ‘Refine box b\*’)*.
>
> A2: Thanks for your detailed comment. This refined box b* is generated by SEPG stage and serves as one part of the prompt, subsequently inputted into PGSR.
>
>
>
> Q3: *The concept ‘proposal seeds’ is proposed in Section 3.1, but its specific generation way (i.e., outer rectangle of the mask generated by SAM) is given until Section 3.4, it is suggested to explain its specific concept early on to avoid confusion. Furthermore, Section 3.4 should be repositioned closer to the beginning of the paper to provide readers with a clearer understanding of the overall method's workflow.*
>
> A3: Thanks for your suggestion. We have reorganized Sections 3.1 and 3.4 in the revised version.
>
>
>
> Q4: *What’s the concrete content of ‘sharing some of the weights’ about the two refiners mentioned in Section 3.2?*
>
> A4: The two refiners involve a shared backbone and, additionally, in the head, there are two shared fully connected (FC) layers. In the revised version, we have added additional details in the Appendix.
>
>
>
> Q5: *What are the key differences in the workflow of the proposed Refiner2 compared to the refining stage in previous works [1]?*
>
> A5: The main differences lies in:
>
> 1) **Structural differences**: The refiner in P2P comprises a classification head and an embedding head, which calculate classification scores and feature embeddings, respectively. The classification scores are utilized for optimizing proposals, while the feature embeddings are employed for updating semantic prototypes. On the other hand, P2BNet utilizes a multi-instance learning structure with two classification heads for computing classification scores and instance scores.
> 2) **Optimization differences**: We use the proposal scores from *Refiner II* as indicators to update the semantic prototypes with high-quality embeddings. The high-quality prototypes are then used in *Refiner I* to compute instance probabilities, obtaining high-quality proposals. When these proposals are input into Refiner 2, it further enhances the quality of semantic prototypes. We believe there is a mutually reinforcing relationship between *Refiner I* and *Refiner II*, which is absent in [1].
>
>
> ------
> *[1]* *Chen, Pengfei, et al. "Point-to-box network for accurate object detection via single point supervision." European Conference on Computer Vision, 2022.*
>
> ------
>
>
> Q6: *How about the computational complexity of the proposed method compared to other methods?*
>
> A6: As suggested, we added a comparison of memory and time consumption in the appendix, and the explanation for this is provided in the *unified response* part.
>
>
>
> Q7: *In Section 3.4, the actual input of the subsequent P2P refiner is the initial proposal seed box (i.e., outer rectangle of the mask generated by SAM). If possible, it is recommended to use the box (e.g., the fully supervised bounding box label) as the prompt of SAM to conduct a more comprehensive experimental comparison in segmentation, rather than just using the point as the prompt of SAM for comparison.*
>
> A7: In the revised version, we supplemented the results using GT bounding box as a prompt in Tab. 3, which can be considered an upper bound for point supervision methods. Using only point as supervision, we reached performance levels of **85% (26.4 vs 31.1)**, **91% (48.6 vs 53.1)**, and **80% (26.2 vs 32.6)** on three respective metrics when using bounding box prompts. This indicates that P2P approaches the performance of bounding box supervision, reaffirming the effectiveness of our method.
>
> | Method | Sup. |AP | AP50 | AP75 |
> | :-------------: | :-----------: | :---------: | :--------------: | :--------------: |
> |     SAM-MR      |     Point     |    24.3     |       43.8       |       24.3       |
> |     SAM-MR      |      box      |    31.1     |       53.1       |       32.6       |
> |     Ours-MR     |     Point     |    26.4     |       48.6       |       26.2       |

---

> ### Author Response · Authors · 2023-11-22
> **Response to reviewer iqSb (2st)**
>
> Dear Reviewer iqSb,
>
> We would like to express our sincere gratitude for your time and efforts in improving our paper.
>
> Considering your concerns about our paper, we have revised some of the content again and summarized it below:
>
> - We've reorganized the methods section, merging 3.4 into 3.1 to give the reader a better understanding of our framework.
>
> - Regarding *Prompt refiner II*, we placed the homogenization with the previous method in the Appendix to highlight the innovations.
>
> We hope the above improvement could clarify your concerns. Looking forward to your feedback and discussion. Thanks again!
>
> Best,
>
> Paper 1568 Authors.

---

### Official Review · Reviewer_7jkG · 2023-10-31

**Soundness:** 3 good
**Presentation:** 3 good
**Contribution:** 3 good
**Rating:** 6
**Confidence:** 5

**Summary:**

The paper introduce foundation model SAM into point supervised object detection and segmentation task. Benefit from the high generalized ability of SAM, the object segmentation map can be easier obtained with the point prompt. However, the mask provided by SAM are not what we need sometimes because SAM is semantic-free and brings ambiguity. The author proposed a method to generate better-quality segmentation map as the pseudo label for training of detection segmentation task. I support for what the author said: ‘rather than directly designing large foundation models, it is more meaningful to leverage them for specific tasks in resource-constrained situations.’

**Strengths:**

I support for the combination of foundation model and specific tasks and interested in the point-supervised tasks.
The proposed method bridges the gap between PSOD (or PSIS) and fully-supervised method.

**Weaknesses:**

There are some questions here:
1、The author said ‘we observe that only 40% of the masks covered more than 70% foreground pixels’. However, as I know, because the semantic-free of SAM, the highest score mask generated by SAM may not what we what, but the top-3 masks may contains what we want in most situation. What we should do is to select the best one (or with extra refinement) from the top-3 masks. But the paper only choose the highest score one and claimed ‘only 40% of the masks covered more than 70% foreground pixels’, I think this is a handmade problem. In other words, the author do not make full use of foundation model and make the problem more difficult.
2、I think the framework is lack of novelty and a little engineering. If I understand you correctly， the structure is (SAM+P2BNet(CBP stage + PBR stage))+(SAM+ P2BNet)+ (SAM+ P2BNet) ....... The whole paragraph of ‘SEMANTIC-EXPLICIT PROMPT GENERATION’ is P2BNet (the network, the loss, the sampling is similar) and the ‘PROMPT GUIDED SPATIAL REFINEMENT’ is the application of SAM. I think the combination of P2BNet and SAM is OK, but the author did not propose some insights or other challenges in combination. And the ITERATIVE LEARNING is a little engineering, I am interested in the time cost in practical application.
3、Some other problems: (1) The visualization is unclear if I do not enlarge the image or I print it. The line is too thin. (2) And I think the title is not suitable because another paper is named as P2P: Rethinking Counting and Localization in Crowds:A Purely Point-Based Framework, ICCV2021. The tasks are different but relevant, and are all point-based. (3) I think the experiments on better detectors or segmentation network (or better backbone?) are needed. The faster RCNN is classic but too old. The fully-supervised method is far beyond 30+ or 40+ on COCO.

**Questions:**

See weakness.

---

> ### Author Response · Authors · 2023-11-20
> **Response to reviewer 7jkG (1/2)**
>
> Thanks for your constructive comments. We hope our responses can address your concerns. Further comments are welcomed.
>
> ------
>
> Q1: *“The author said ‘we observe that only 40% of the masks covered more than 70% foreground pixels’. However, as I know, because the semantic-free of SAM, the highest score mask generated by SAM may not what we what, but the top-3 masks may contain what we want in most situation. What we should do is to select the best one (or with extra refinement) from the top-3 masks. But the paper only chooses the highest score one and claimed ‘only 40% of the masks covered more than 70% foreground pixels’, I think this is a handmade problem. In other words, the author does not make full use of foundation model and make the problem more difficult.”*
>
>
>
> A1: Thank you for your insightful comment. We perceive this as two distinct approaches. To obtain semantically accurate masks, as you mentioned, one could design a selector to choose the most accurate mask from the three. This might yield better results, but it does not guarantee success in complex scenarios.
>
> We analysis that the significant semantic differences in masks generated by SAM can be attributed fundamentally to the semantic uncertainty in the provided prompts. Thus, we can address this by improving the quality of the prompts themselves. By progressively enhancing the semantic confidence of the prompts, we aim to ensure the foundation model outputs accurate masks.
>
>
>
> Q2: *“I think the framework is lack of novelty and a little engineering. If I understand you correctly， the structure is (SAM+P2BNet(CBP stage + PBR stage))+(SAM+ P2BNet)+ (SAM+ P2BNet) ....... The whole paragraph of ‘SEMANTIC-EXPLICIT PROMPT GENERATION’ is P2BNet (the network, the loss, the sampling is similar) and the ‘PROMPT GUIDED SPATIAL REFINEMENT’ is the application of SAM. I think the combination of P2BNet and SAM is OK, but the author did not propose some insights or other challenges in combination. And the ITERATIVE LEARNING is a little engineering, I am interested in the time cost in practical application.”*
>
> A2:
>
> 1. Firstly, the *main distinctions between SEPG and P2BNet* are as follows:
>
>    - **differences in Structure**: Diverging from classical MIL frameworks, we employ a more stable feature prototype for computing instance-level probabilities. The refiner in P2P comprises a classification head and an embedding head, which calculate classification scores and feature embeddings, respectively. P2BNet utilizes the MIL structure with two classification heads for computing classification scores and instance scores.
>
>    - **Differences in Sampling**: In order to mitigate the randomness associated with sampling and optimization, we employ a **seed-based group sampling** strategy, leveraging prior knowledge provided by the foundation model. In contrast, P2BNet employs point-centered neighbor sampling.
>
>    - **Differences in optimization**: P2P follows a **group-then-individual** fashion to reduce the solution space. In Prompt refiner I, semantic and instance probabilities for each group are computed, and through optimizing the joint probability, the group with the highest score is selected. And then, the proposals in the selected group are augmented as the input of the Prompt refiner II. P2BNet employs a multi-level individual optimization approach.
> 2. Furthermore, we would like to emphasize that our novelty extends beyond individual components. Our work contributes novel insights to the community in addressing weakly supervised or point-supervised tasks, specifically focusing on how to better leverage foundation models in the era of the foundation model to improve solutions for weakly supervised problems.
> 3. Regarding time consumption, while P2P consumes a bit higher time and memory during the training phase, it yields pseudo-labels with higher quality. We think that the time consumption of point-supervised methods itself may not be a primary limitation in practical applications. Leveraging these high-quality pseudo-labels, it is free for users to employ any detectors to conduct specific tasks or applications. Therefore, compared to the substantial improvement in performance, the time consumption during training is not an issue.

---

> > ### Comment · Reviewer_7jkG · 2023-11-21
> >
> > R to A1: Yep, I agreed with the motivation to progressively enhance the semantic confidence of the prompts for better SAM result. I thought about it，and I agreed that think they are two approaches that do not conflict. So, I want to let them as the advice that select the best proposal from the three will give better initialisation for your approach. We hope this will bring you some insight.
> >
> > R to A2: First, I agree with that the novelty is how to make full use of the ability of foundation model in weakly-supervised task, which extends beyond individual components. In original review, the homogeneity with P2BNet results in low score. In rebuttal, I think the author make it clear in A2-1 and address my concern.  I hope the author will reorganise the methods section, simplify the same and highlight the differences. Due to this will not affect the main contribution,  I will raise my score.
> >
> > R to A3：Because in inference stage, only the retrained detector or segmentation model will be used, so the inference time depends on the retrained network. I am still interested in the implement details in  ITERATIVE LEARNING，which will affect the training time I think. Do you extract the image feature wil SAM and save it in memory first, then you conduct your method and generate better prompt for SAM and use the prompt to produce better mask in mask decoder of SAM, finally do the interative learning? So you can make it clear that which component will be repeatly used in interative refinement and which will be conduct only one time.

---

> ### Author Response · Authors · 2023-11-20
> **Response to reviewer 7jkG (2/2)**
>
> Q3.1: *“The visualization is unclear if I do not enlarge the image or I print it. The line is too thin.”*
>
> A3.1: Thank you for your reminder. In the revised version, we have redrawn the visualization to address the concerns.
>
>
>
> Q3.2: *“And I think the title is not suitable because another paper is named as P2P: Rethinking Counting and Localization in Crowds: A Purely Point-Based Framework, ICCV2021. The tasks are different but relevant, and are all point-based.”*
>
> A3.2: Thanks for the comment. The paper you referenced presents a point-to-point network named **P2PNet** designed for crowd counting tasks. While both employ the term *P2P*, the tasks and expressions hold distinct meanings. Our reference to "point to prompt" reflects a different context. Hence, we consider this to be coincidental. We appreciate your feedback and are open to making adjustments if necessary.
>
>
> ------
> [1] *Song, Qingyu, et al. "Rethinking counting and localization in crowds: A purely point-based framework." Proceedings of the IEEE/CVF International Conference on Computer Vision. 2021.*
>
> ------
>
> Q3.3: *I think the experiments on better detectors or segmentation network (or better backbone?) are needed. The faster RCNN is classic but too old. The fully-supervised method is far beyond 30+ or 40+ on COCO*.
>
> A3.3: Thank you for your insightful advice. To ensure a fair comparison with previous methods, we refrained from employing more advanced detectors or segmentation networks. In the revised version, as suggested, we have supplemented the experiments in the appendix on advanced detectors and segmentation networks.
>
> 1) When **DINO (both R-50 and Swin-L are used as backbone)** [2] is adopted as the detector, our method achieves a 38.2 AP (R-50) and 45.1 AP (Swin-l) on COCO dataset, approximately 80% of the fully supervised performance, and a 57.2 AP50 (R-50) and 66.1 AP50 (Swin-l), about 87% of the fully supervised performance.
>
> 2) When the **Mask2Former** [3] is employed as the segmentation network, achieving a 34.9 AP and 58.9 AP50.
>
> Due to the limited time, we do not adjust the parameters carefully, but it is sufficient to verify the effectiveness of the proposed method. We believe that the performance will be further improved. In summary, when utilizing more advanced detector or segmentation network, our method still closely approaches the performance of fully supervised methods.
>
> |   Method    | backbone |  AP  | AP50 | AP75 |
> | :---------: | :------: | :--: | :--: | :--: |
> |    DINO     |   R-50   | 49.0 | 66.4 | 55.3 |
> |    DINO     |  Swin-l  | 57.2 | 75.7 | 62.7 |
> |  Ours-DINO  |   R-50   | 38.2 | 57.2 | 40.9 |
> |  Ours-DINO  |  Swin-l  | 45.1 | 66.1 | 48.9 |
> | Mask2former |  Swin-s  | 46.1 | 69.4 | 49.8 |
> |   Ours-MF   |  Swin-s  | 34.9 | 58.9 | 37.6 |
>
>
> ------
> *[2] Zhang, Hao, et al. "DINO: DETR with Improved DeNoising Anchor Boxes for End-to-End Object Detection." The Eleventh International Conference on Learning Representations. 2022.*
>
> *[3] Cheng, Bowen, et al. "Masked-attention mask transformer for universal image segmentation." Proceedings of the IEEE/CVF conference on computer vision and pattern recognition. 2022.*

---

> > ### Comment · Reviewer_7jkG · 2023-11-21
> >
> > R to A3.2. Because they are exactly the same, so the title depends on you. I just tell you that I have read a paper with a similar abbreviation of your method (P2PNet vs. P2P)
> >
> > R to A3.3, I am satisfied with the experiments on sota detector and segmentation method. Add it to the table in the paper. By the way, please mark the training details (e.g. the training epoch (2P and retained, respectively), the backbone (P2P and retained, respectively) and others you think important). And, please add a swin-l mask2former when you have time afterwards.

---

> ### Author Response · Authors · 2023-11-22
> **Response to reviewer 7jkG (2st)**
>
> 1. Thank you very much for your suggestion, we strongly agree that this is a good idea and inspired us a lot. We will try it if time permits.
>
> 2. We are encouraged that your concerns about the novelty of our approach have been addressed and thank you for your positive comments.  We have reorganized the methods part following your advice in the latest revised version.
>
> 3. In the iterative learning part, each iteration consists of (SEPG+PGSR), we first train SEPG for 12 (when T=1) or 6 (when T > 1) epochs to get a semantically clear prompt, and then input it into SAM as PGSR stage and output a better mask. if T=1, the iteration is terminated, if T>1, we continue to transform the obtained mask into a proposal seed for the next round of iteration.
> Instead of pre-saving the features, we re-extract the features in every iteration.
>
> We strongly agree that it is a good method to reduce the time cost  by pre-saving the features into memory. This won't make a difference to the results, but it will save time for SAM to extract the features. We will try to apply this idea.
>
> Thank you again for your time and effort in improving our work. Further comments are welcomed.

---

> ### Author Response · Authors · 2023-11-22
> **Response to reviewer 7jkG (2st)**
>
> We added the number of retrained epochs in the table, following the default settings. And we have been added detailed explanations to the revised version. Thanks for your advice.
>
>
> |   Method    | Backbone | Retrained Epoch |  AP  | AP50 | AP75 |
> | :---------: | :------: | :-------------: | :--: | :--: | :--: |
> |    DINO     |   R-50   |       12e       | 49.0 | 66.4 | 55.3 |
> |    DINO     |  Swin-l  |       12e       | 57.2 | 75.7 | 62.7 |
> |  Ours-DINO  |   R-50   |       12e       | 38.2 | 57.2 | 40.9 |
> |  Ours-DINO  |  Swin-l  |       12e       | 45.1 | 66.1 | 48.9 |
> | Mask2former |  Swin-s  |       50e       | 46.1 | 69.4 | 49.8 |
> |   Ours-MF   |  Swin-s  |       50e       | 34.9 | 58.9 | 37.6 |

---

### Author Response · Authors · 2023-11-20
**Unified Response**

Reviewers expressed considerable concern about the practical complexity of our method, particularly regarding time and computational resource consumption. Here, we aim to provide a unified response to address this issue.

we have supplemented detailed comparisons of memory, training time, and inference time in the appendix of revised version. The results can also be observed from the tables below.

We first clarify the workflow of point supervision methods: in practical applications, point supervised methods involves initially training a  regressor to obtain high-quality pseudo-labels. Subsequently, using these pseudo-labels as supervision, one can choose an appropriate subnetwork (*e.g.*, a detector ) for specific tasks or applications and opimized it in a fully supervised manner with the predicted pseudo labels,  (*e.g.*, we can use more powerful detectors, such as DINO, to ensure performance, or choose faster detectors like YOLO to guarantee real-time capabilities.).

While P2P consumes a bit higher time and memory during the training phase, it yields pseudo-labels with higher quality. We think that the time consumption of point-supervised methods itself may not be a primary limitation in practical applications. Leveraging these high-quality pseudo-labels, it is free for users to employ any detectors or segmentation networks to conduct specific tasks or applications. Therefore, compared to the substantial improvement in performance, the time consumption during training is not an issue.



| Method | Model |  Epoch  | Mem. (M) | t/Epoch(h) | Inf. time (s) |
| :----: | :---: | :-----: | :------: | :--------: | :-----------: |
| P2BNet | R-50  |   12    |   4074   |   ~1.08    |     0.078     |
|  SEPG  | R-50  | 12 or 6 |   4598   |   ~1.25    |     0.090     |
|  PGSR  | ViT-H |    -    |   7696   |   ~2.4h    |     0.292     |

|  T   | Epochs/iter | Total time (h) |
| :--: | :---------: | :------------: |
|  1   |     12      |      ~17       |
|  2   |      6      |      ~20       |
|  3   |      6      |      ~30       |
|  4   |      6      |      ~40       |

---

### Author Response · Authors · 2023-11-20
**Response to all reviewers**

We appreciate all the reviewers for their valuable and constructive comments. All reviewers acknowledged the significance of our method and its surpassing performance. Specifically, *Reviewer 7jKG* highlighted that our method "bridges the gap between PSOD (or PSIS) and fully-supervised methods." *Reviewers iqSb* and *6Xmh* appreciated the logic and writing of our paper, while *Reviewer j4vZ* praised our motivation as "reasonable and easy to understand."

We respond to the reviewers' comments one by one. And we have revised our paper accordingly and summarized the main improvements below:

1. Add experiments on better detector and segmentation network. (by Reviewer 7jkG)
2. Redraw the visualizations. (by Reviewer 7jkG)
3. Added experiments using bounding boxes as prompts. (by Reviewer iqSb)
4. Performed a more detailed comparison of performance with SAM. (by Reviewer j4vZ)
5. Compared and analyzed the consumption of time and computational resources. (by Reviewer 6Xmh and j4vZ)

---

### Meta-Review · Area_Chair_U7Lz · 2023-12-05

**Metareview:**

The paper presents a promising approach in point-supervised object detection and segmentation, leveraging the capabilities of a foundation model. The novel iterative learning framework and the method's performance on multiple datasets are notable strengths. However, concerns about the limitations of the foundation model, lack of novelty in some aspects, and computational demands are significant weaknesses. The mixed reviews place the paper at the borderline. The AC checked all the reviews and discussions, and believe the major concerns raised by the reviewers are valid. Thus, the paper is rejected.

**Justification For Why Not Higher Score:**

Concerns about the limitations of the foundation model, lack of novelty in some aspects, and computational demands are significant weaknesses.

**Justification For Why Not Lower Score:**

N/A

---

### Decision · Program_Chairs · 2024-01-16

Reject